# Scaling Infrastructure to Support Multi-Trillion Parameter LLM Training

Mikhail Isaev[*], Nic McDonald[†], and Richard Vuduc[*]

[*]Georgia Institute of Technology, Atlanta, GA, USA

[†]NVIDIA, Salt Lake City, UT, USA

*Abstract*—This paper discusses efficient system designs for Large Language Model (LLM) scaling to up to 128 trillion parameters. We use a comprehensive analytical performance model to analyze how such models could be trained on current systems while maintaining 75% Model FLOPS Utilization (MFU). We first show how tensor offloading alone can be used to dramatically increase the size of trainable LLMs. We analyze performance bottlenecks when scaling on systems up to 16,384 GPUs and with models up to 128T parameters. Our findings suggest that current H100 GPUs with 80 GiB of HBM enabled with 512 GiB of tensor offloading capacity allows scaling to 11T-parameter LLMs; and getting to 128T parameters requires 120 GiB of HBM and 2 TiB of offloading memory, yielding 75%+ MFU, which is uncommon even when training much smaller LLMs today.

## I. Introduction

We wish to consider what software and system configurations might permit existing Large Language Models (LLMs), now at about 1 trillion parameters [11], to scale with greater efficiency to even larger model sizes. Our analysis is driven by the continued success and efficacy of LLMs in a variety of applications [2], [3], [7], [11], [14], [16], [21] and motivated by the observation that Model FLOPS Utilization (MFU)—a common metric of efficiency for assessing how well specialized Artificial Intelligence (AI) accelerators are utilized during model training—can be as low as 50% or less [15].

A significant improvement to MFU will be necessary to increase model sizes by $10\times$ (10 trillion parameters) or higher on architectures similar to current systems. With a space requirement of 20 bytes per parameter, to store just the model's weights and optimizer state we would need more than 200 TB of memory. For a system based on NVIDIA H100 [12] Graphics Processing Unit (GPU) with 80 GiB of high bandwidth memory (HBM) memory, we would need 2,500 GPUs and a fully model-parallel implementation to train such a model. No known model-parallelism technique at this scale would be able to provide anywhere near 50% MFU.

Motivated by this example, we aim to establish the system limitations that prevent us from training multi-trillion parameter models on large systems built using clusters of 8 interconnected GPUs, similar to NVIDIA DGX and HGX. We start by presenting a methodology for choosing well structured multi-trillion parameter LLMs. Then, using our own fast analytical performance model of transformer-based LLM training, we search a space of billions of system configurations and execution strategies. This paper explains a few of our findings, which may be summarized as follows.

1) Training a hundred-trillion parameter LLM is feasible but requires a secondary memory pool up to 1 TiB per GPU with a bandwidth of 100 GB/s bidirectionally.
2) Strong scaling for a 1T model stalls around 12,288 GPUs, as matrix multiply becomes small, inefficient, and unable to overlap with communication.
3) Scaling beyond 10T models requires more first-level memory, with HBM size scaling with model size.
4) Growing model and system size beyond 10T parameters and 10k GPUs demands a larger fast-network domain and more targeted software optimizations.

Overall, we find it will be critical to co-design the LLM, software, and hardware to attain high performance and efficiency.

## II. Experiments Methodology

For performance estimation we use Calculon [5], a fast open source analytical model of LLM training performance that we developed.[1] Calculon can estimate the time and resource usage for a given LLM, system configuration, and software execution strategy in about 1 millisecond, allowing the exploration of large design spaces having many billions of such configurations. Calculon models LLM training with tensor parallelism (TP), pipeline parallelism (PP), and data parallelism (DP), allowing searches to determine optimal split-parallelism configurations. The system specification describes an accelerator-based distributed system with a two-level memory hierarchy connected to multiple networks shown on Fig. 1.

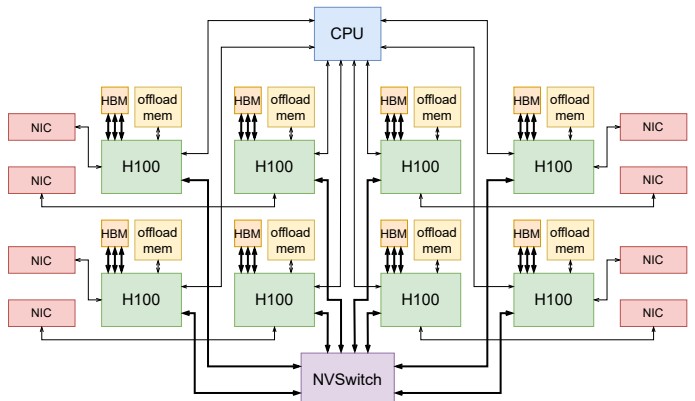

Fig. 1. Node architecture used for system modeling.

[1]The full description of Calculon will be available in a future paper.

To validate the accuracy of its modeling, Calculon was compared against actual runs of Megatron LLM on NVIDIA's A100-based Selene supercomputer [8]. Calculon achieves a high level of accuracy, with an average error of 3.4% and a maximum error of 7.25% on these validation runs as presenetd in Table I.

TABLE I
VALIDATION RESULTS COMPARING CALCULON'S PERFORMANCE PREDICTION TO ACTUAL RUNS ON THE A100-BASED SELENE [8]

|  |  | 22B | 175B | 530B | 1T |
|---|---|---|---|---|---|
| *Full* | **Selene** | 1.42 | 18.13 | 49.05 | 94.42 |
|  | **Calculon** | 1.43 | 18.30 | 50.46 | 91.70 |
|  | **Delta** | -0.40% | -0.94% | -2.88% | -2.88% |
| *Seq+Sel* | **Selene** | 1.10 | 13.75 | 37.83 | 71.49 |
|  | **Calculon** | 1.17 | 13.92 | 35.09 | 67.74 |
|  | **Delta** | -6.36% | -1.24% | 7.25% | 5.24% |

We perform experiments that vary system size, model size, memory capacity, bandwidth, and NVLink domain sizes, working with the FP8 data format supported by H100. For each system, we pick an execution strategy that considers multiple state-of-the-art software optimizations [8], [11], [17]–[19] and picks the best-performing one. Given the large search spaces, we cannot present our experiments fully and instead focus on a few of the most important trends we have discovered.

Our analysis assumes a networked system of compute nodes whose node-architecture is depicted in Fig. 1. It is similar to DGX or HGX in structure and connectivity. The only difference is the addition of offload memory attached to GPU in addition to HBM. Such memory can be connected via compute express link (CXL), or hosted by Central Processing Unit (CPU) and made directly accessible from GPU, similar to NVIDIA's Grace-Hopper [13].

## III. SELECTION OF LLM CONFIGURATIONS

An important parameter is the LLM's *aspect ratio*, defined as the ratio between the hidden dimension of the transformer block to the number of blocks (a.k.a., transformer layers). Some recent research claims the ideal aspect ratio is a constant 128 [6], while others claim that the aspect ratio should increase exponentially with the number of blocks [9]. Both of these analyses were performed on LLMs 2 to 5 orders of magnitude smaller than today's production LLMs. In the absence of consensus among the LLM experts, we follow the apparent current practice suggested by Table II, which is to extrapolate aspect ratios linearly with the number of transformer blocks. Nevertheless, our analysis method would work for any scaling function.

In scaling and shaping the LLM, one challenge is mapping the models onto the available hardware. Some models, such as GPT-3 [2] with its 175 billion parameters across 96 blocks, are designed with many dimensions as powers of two or multiples of powers of two, making them well-suited to typical system designs that are also commonly built in powers of two. Other models are not as easy to map. Turing-NLG [20] has 530 billion parameters across 105 blocks, which results in

TABLE II
ASPECT RATIOS OF CURRENT LLMs.

| Name | Hidden | # Blocks | Aspect Ratio |
|---|---|---|---|
| GPT2-1.5B [16] | 1600 | 48 | 33.3 |
| Jurassic-6.5B [10] | 4096 | 32 | 128 |
| PaLM-8B [3] | 4096 | 32 | 128 |
| GPT3-13B [2] | 5140 | 40 | 128.5 |
| Megatron-40B [11] | 6144 | 40 | 153.6 |
| PaLM-62B [3] | 8192 | 64 | 128 |
| Chinchilla-64B [4] | 8192 | 80 | 102.4 |
| GPT3-175B [2] | 12288 | 96 | 128 |
| Jurassic-175B [10] | 13824 | 76 | 181.9 |
| Megatron-309B [11] | 16384 | 96 | 170.7 |
| TuringNLG-530B [20] | 20480 | 105 | 195 |
| PaLM-540B [3] | 18432 | 118 | 156 |
| Megatron-1T [11] | 25600 | 128 | 200 |

fewer possible mappings. PaLM [3] has 540 billion parameters across 118 blocks, a prime number multiplied by 2, which results in even fewer.

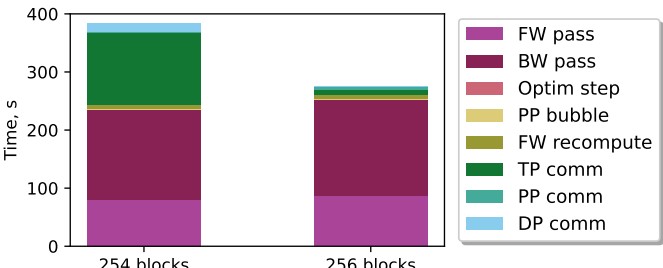

Fig. 2. Performance comparison of two 11T models.

To see the impact of such choices, Fig. 2 compares two similarly sized models of about 11 trillion parameters. One has a power of two number of blocks (256) and the other has a prime number multiplied by two (254). When mapped onto 4,096 processors, the 256-block model yields 15,612,832 possible mappings while the 254-block model yields only 842,080, or 18.5× fewer. Consequently, the 256-block model ends up being 36% faster, with an MFU of 75% compared to the 254-block model's MFU of 54%.

Thus, we propose scaling the number of blocks and attention heads with a step size that is a power of two. Doing so makes it easier to configure tensor and pipeline parallelism, yielding better overall performance. Fig. 3 summarizes the model sizes for a variety of aspect ratios. These models all result in many millions of mapping solutions on various common system designs and across many system sizes.

The hidden step size shown in Fig. 3 is 8,192. However, when finding the optimal (closest to ideal aspect ratio), we use a step size of 1,024. For the remainder of this paper we use the model configurations found in Table III. All models have a sequence size of 8,192, the feed forward size is fixed to 4× the hidden size, and the number of attention heads is equal to the number of blocks. For all experiments we limited the maximum batch size to 3,072.

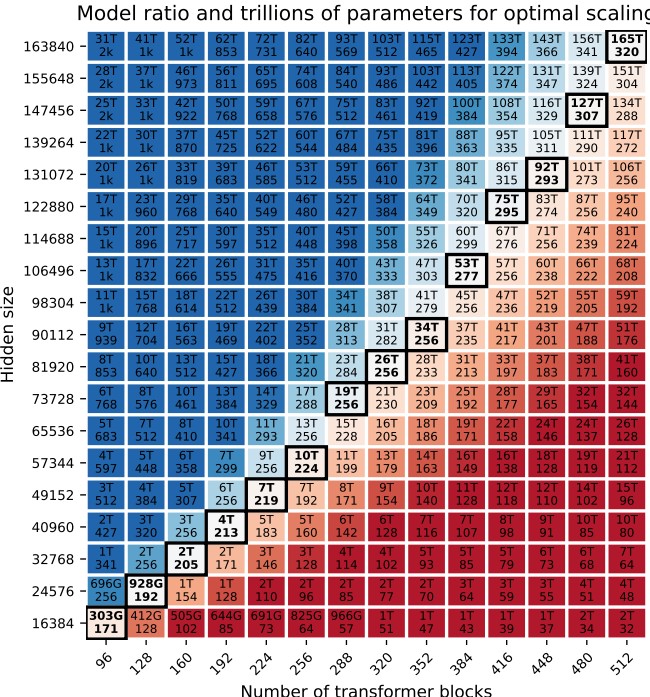

Fig. 3. Linear scaling of the hidden size with number of transformer blocks, in steps of 8,192 for hidden size and 32 for number of blocks. Each cell contains model size and hidden to blocks ratio. Red color represents narrower models, blue color represents wider ones. Optimal choices are represented by white color in the frame, model size and ratio in bold.

TABLE III
TWELVE MULTI-TRILLION PARAMETER LLMS, FROM 1T TO 128T.

| Name | Hidden | Attn Size | # Blocks | Aspect Ratio |
|------|--------|-----------|----------|--------------|
| 1T | 24,576 | 192 | 128 | 192 |
| 2T | 32,768 | 205 | 160 | 204.8 |
| 4T | 40,960 | 213 | 192 | 213.3 |
| 7T | 50,176 | 224 | 224 | 224 |
| 11T | 60,416 | 236 | 256 | 236 |
| 18T | 70,656 | 245 | 288 | 245 |
| 26T | 81,920 | 256 | 320 | 256 |
| 37T | 94,208 | 268 | 352 | 267.6 |
| 53T | 106,496 | 277 | 384 | 277.3 |
| 72T | 119,808 | 288 | 416 | 288 |
| 96T | 134,144 | 299 | 448 | 299.4 |
| 128T | 148,480 | 309 | 480 | 309.3 |

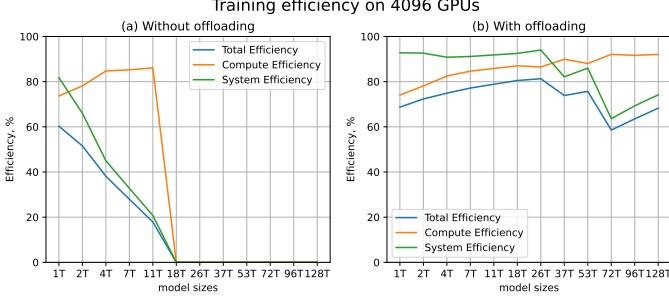

Fig. 4. Comparison of LLM scaling on 4,096 GPUs with and without offload memory. Such memory enables high training efficiency beyond 100T models.

## IV. TENSOR OFFLOADING FOR LLM SCALING

While scaling out LLMs using standard DGX/HGX H100s with 8 NVLink-connected GPUs is possible, achieving high performance is not trivial. See, for instance, Fig. 4a, which shows training efficiency while scaling up model size on a fixed system size of 4,096 GPUs. Even the smallest model size, 1T, reaches only 60% efficiency and rapidly decays until 18T where it can no longer run. The main scalability issue is the lack of memory to store weights and activations during training. This in turn forces the use of activation recomputation and higher model parallelism. A large pipeline parallelism with a lack of spare memory forces an excessive time overhead in the form of a pipeline bubble. A large tensor parallelism beyond the NVLink size of 8 increases communication time due to a lack of bandwidth.

These issues can be addressed by a secondary memory pool, where unused tensors from inactive transformer blocks can be transferred and retrieved as needed [19]. This could be implemented as CPU host memory, an array of PCIe-attached SSDs, or CXL-attached memory. We consider training efficiency when using tensor offloading in Fig. 4b, where the per-GPU capacity is 1 TiB at infinite bandwidth. Evidently, with enough offloading capacity and infinite offloading bandwidth, we could train models at least up to 128T parameters.

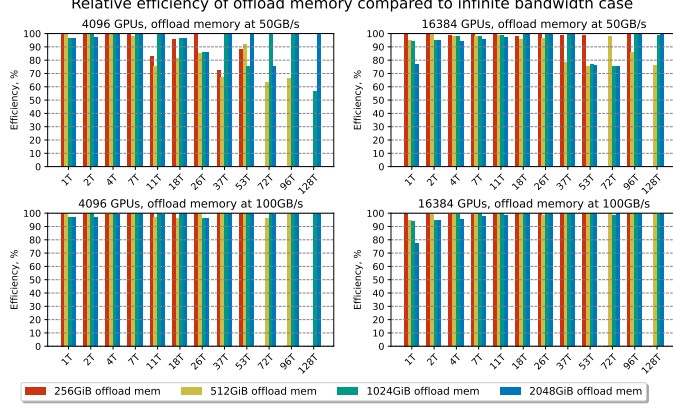

Fig. 5. Efficiency of offload memory compared to infinite offload bandwidth.

The effect of offloading capacity is compared in Fig. 5 for 256 GiB, 512 GiB, 1 TiB, and 2 TiB. We see the relative slowdown of using 50 GB/s and 100 GB/s of offloading bandwidth per direction compared to infinite bandwidth. At 50 GB/s on a 4,096 GPUs system, significant slowdowns occur with increasing model sizes. At 100 GB/s, the majority of the systems nearly match the performance of infinite bandwidth, suggesting it is a sufficient target bandwidth.

Importantly, these tensor offload-memory requirements are within reach of current technology. Memory pools based on CXL 1.1 and CXL 2.0 with a capacity up to 2 TiB and bandwidth up to 89.6 GB/s are already available [1]. Systems based on NVIDIA's Grace-Hopper [13] have up to 512 GiB of Low-Power Double Data Rate (LPDDR) memory with up to 546 GB/s bandwidth behind a CPU-to-GPU link, far above our offloading-requirement estimates.

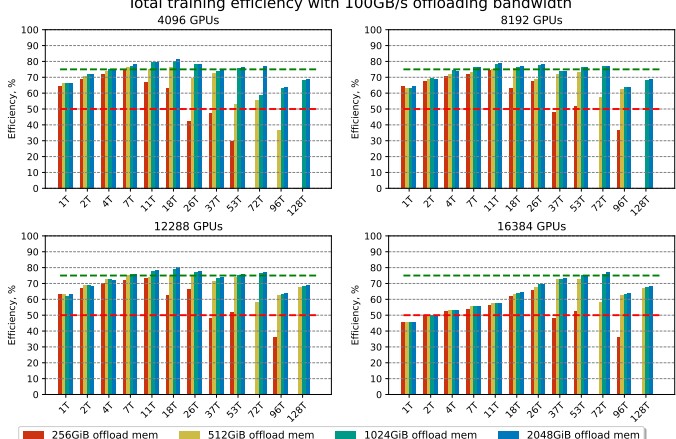

Fig. 6. Training efficiency with model and system scaling using offloading memory with infinite bandwidth. Green dash line indicates 75% MFU, red dash line indicates 50% MFU.

The efficiency of training with offloading appears in Fig. 6 for an offload bandwidth of 100 GB/s and capacities of 256 GiB, 512 GiB, 1 TiB, and 2 TiB across 4,096, 8,192, 12,288, and 16,384 GPUs. The major trends shown are:

- Small models on large systems and large models on small systems lead to low efficiency.
- Good efficiency occurs rarely at 256 GiB.
- For 8k, 12k, and 16k GPUs, 512 GiB is mostly sufficient.
- A 1 TiB capacity is nearly identical to 2 TiB.

## V. Strong Scaling

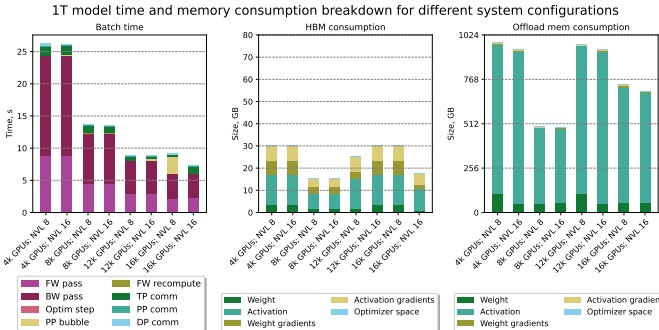

Fig. 7. Batch time and memory consumption break down for 1T model strong scaling from 4,096 to 16,384 GPUs.

In this section we analyze the strong scaling of the 1T parameter model from 4,096 to 16,384 GPUs inspecting NVLink domain sizes of 8 and 16. For each system, we use Calculon to perform an exhaustive search over possible configurations, typically 10-30 million configurations per LLM-system pair. The results appear in Fig. 7. We analyzed all configuration associated parameters such as TP, PP, DP split, microbatch size, pipeline interleaving, among others, and summarize notable trends below.

Scaling up to 12,288 GPUs fares well but suffers at 16,384 GPUs. An NVLink size of 8 is sufficient up to 12,228 GPUs but 16 is needed for higher efficiency at 16,384 GPUs. Adding

extra processors requires assigning them to tensor, pipeline, or data parallelism, but each incurs some resource cost in time or memory. We identified several reasons for a lack of scaling at 16,384 GPUs.

1) When increasing TP the tensor may be divided too finely to maintain a high compute efficiency on the GPU.
2) When increasing TP the size of each message may become small enough to become latency dominated.
3) When attempting to overlap TP communication and computation, increasing TP reduces the computation size but communication size remains the same. At particular FLOPs/bandwidth ratios, the communication-hiding decreases, reducing efficiency.
4) When overlapping TP communication and computation, to sustain the high bandwidth of NVLink the GPU must dedicate many cores to communication reducing its computational speed. Adding a specialized direct memory access (DMA)-like engine for communication would eliminate this overhead allowing optimal overlap.
5) Increasing PP either increases the pipeline bubble overhead or requires more memory for higher levels of interleaving to reduce the pipeline bubble.
6) Increasing DP increases memory due to replication.
7) We constrain our models to have a maximum batch size of 3,072 to conserve the convergence properties of prior studies. This choice limits the maximum available DP to 3,072, so that the rest must be either TP or PP.

## VI. Scaling Models Beyond 10T Parameters

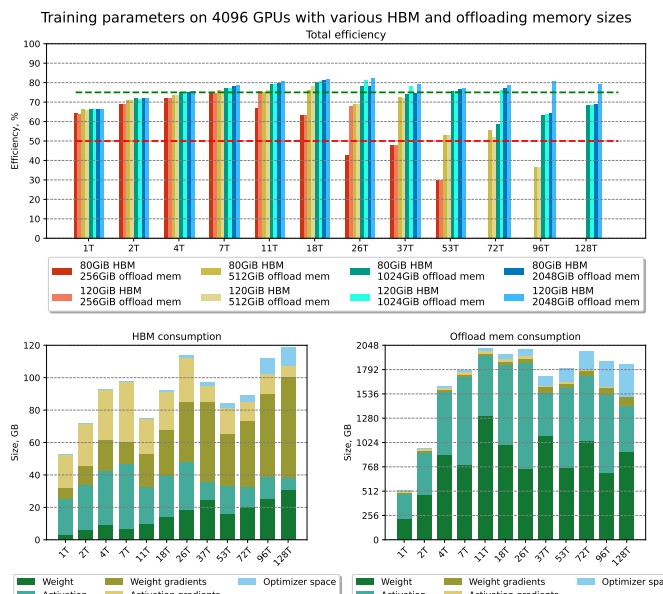

Fig. 8. Efficiency and memory consumption for LLM training on 4,096 GPUs. Green dash line indicates 75% MFU, red dash line indicates 50%. Memory consumption presented for the 120 GiB HBM and 2 TiB offload memory.

We also analyze the effects of increasing the model size to 128T parameters for a system with a fixed number of GPUs. Fig. 8 shows the results for 4,096 GPUs with 80 GiB and 120 GiB of HBM and 256 GiB, 512 GiB, 1 TiB, and 2 TiB of

offloading capacity. While scaling model training on 4,096 GPUs works well with 80 GiB of HBM for models up to 11T parameters, the HBM size must increase to 120 GiB to scale further, even when given extra offloading memory. The reason is that there must be enough HBM memory to hold two transformer blocks—the one used in computation and the one needed for offloading and prefetching—even with offloading. During model scaling, the transformer block size grows mostly due to weights and activations. Unsurprisingly, offload memory capacity also needs to scale accordingly.

Our experiments indicate that growing the HBM size to 120 GiB and offload memory to 2 TiB is enough to scale to 100T parameters. Past 11T parameter, models occupy most of the available memories. This indicates that further efficiency improvements are possible, either by providing more memory, or by increasing the size of the NVLink domain to reduce per-GPU weight space and increase local microbatch size. These experiments show that the proposed LLMs can scale up to 128T parameters while maintaining an MFU above 75%, which is better than typically seen on current systems for much smaller LLMs.

## VII. CONCLUSION

Our co-design analysis reveals that it is feasible to train a well-structured multi-trillion parameter LLM efficiently at 75% MFU or higher with an appropriate choice of software and hardware settings, including a secondary memory pool for tensor offloading. We identify both optimal configuration strategies and fundamental limitations under strong scaling (fixed model, increasing numbers of GPUs). And for a fixed system with 4,096 GPUs, we show how an 11T parameter model could be trained with only tensor offloading, as well as how to scale to 128T parameters using a 120 GiB HBM and a 2 TiB offloading memory.

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
