# OpenReview forum: "Scaling Infrastructure to Support Multi-Trillion Parameter LLM Training"
_iscaconf.org/ISCA/2023/Workshop/ASSYST — ASSYST Oral_

### Official Review · Reviewer_8GNg · 2023-05-04
**Good exploration but missing details**

**Rating:** 6
**Confidence:** 2

**Review:**

The paper presents a detailed exploration of the system and model sizes to efficiently train the LLMs with multi-trillion parameters. The exploration relies on an analytical model, Calculon. However, the details of calculon are missing.

1. How is the modeling done using calculon and what kind of search space exploration does it do to provide optimal configurations?

2. How is the offload memory modeled in calculon?

3. Is calculon validated with some real hardware or published literature?

4. In the explanation with strong scaling, the explanations start with "when increasing TP". However, there is no mention anywhere what is the degree of PP, TP, DP in all of these experiments. It is hard to deduce what happens when TP is increased. Can you please point out to where in the figure I might be able to see the effect of increasing TP?

**Review (Strengths/Weaknesses):**

1. How is the modeling done using calculon and what kind of search space exploration does it do to provide optimal configurations?

2. How is the offload memory modeled in calculon?

3. Is calculon validated with some real hardware or published literature?

4. In the explanation with strong scaling, the explanations start with "when increasing TP". However, there is no mention anywhere what is the degree of PP, TP, DP in all of these experiments. It is hard to deduce what happens when TP is increased. Can you please point out to where in the figure I might be able to see the effect of increasing TP?

**Reviewer Expertise:**

Knowledgeable: I used to work in this area and/or I try to keep up with the literature but might not know the latest developments.

---

### Official Review · Reviewer_wpwu · 2023-05-05
**Short paper with a lot of great insights!**

**Rating:** 7
**Confidence:** 4

**Review:**

This was a great read. The paper tackles a very timely and important problem and offers many insights. It presents a lot of meaningful scalability studies in only four pages!

**Review (Strengths/Weaknesses):**

Strengths:
* Well written paper that tackles a timely and important problem.
* The methodology of scaling the model aspect ratio and exploring both model, data, and pipeline parallelism is strong.
* Many great insights.

Weaknesses:
* There is no verification of the model in smaller LLMs on today’s hardware.


**Reviewer Expertise:**

Expert: I have written one or more papers on this topic and/or I currently work in this area.

---

### Official Review · Reviewer_qKLq · 2023-05-10
**Scaling Infrastructure to Support Multi-Trillion Parameter LLM Training**

**Rating:** 6
**Confidence:** 3

**Review:**

Summary:
* The paper provides studies about model construction in terms of model aspect ratio and total parameter size for LLM such that large models can be trained with reasonable MFU
* The authors use an in house analytical model based tool called Calculon, which estimates time and resource usage for training of a given LLM.
* The paper proposes scaling the aspect ratio in powers of two to achieve higher MFU through tensor and pipeline parallelism.
* The paper also suggest using a 1TB large tensor memory for each GPU node to enable efficient of models with trillions of parameters

**Review (Strengths/Weaknesses):**

Strengths
* The study of model aspect ratio and MFU is an interesting and the result depict that optimal model parameters lie outside of conventional best practice.
* The study captures a wide range of model sizes scaling up to 128T.

Additional points to improve
* Given the studies are performed using Calculon, which is an in-house tool. I would suggest providing a description of the analytical model, the input-output parameters and the scope of modeling to put the obtained results in context.
* The paper suggest a non-trivial amount of memory ~1TB per node to store intermediate tensors. Adding such amount of system memory is not trivial and would require reasonable architecture changes. It would be helpful for readers and future researchers if the authors provide some details about reasonable approaches to implement such a system.

**Reviewer Expertise:**

Knowledgeable: I used to work in this area and/or I try to keep up with the literature but might not know the latest developments.